# Chronic Endometritis and Antimicrobial Resistance: Towards a Multidrug-Resistant Endometritis? An Expert Opinion

**DOI:** 10.3390/microorganisms13010197

**Published:** 2025-01-17

**Authors:** Francesco Di Gennaro, Giacomo Guido, Luisa Frallonardo, Laura Pennazzi, Miriana Bevilacqua, Pietro Locantore, Amerigo Vitagliano, Annalisa Saracino, Ettore Cicinelli

**Affiliations:** 1Clinic of Infectious Diseases, Department of Precision and Regenerative Medicine and Ionian Area (DiMePRe-J), University of Bari “Aldo Moro”, Piazza Giulio Cesare n. 11, Cap 70124 Bari, Italy; francesco.digennaro1@uniba.it (F.D.G.); giacguido@gmail.com (G.G.); annalisa.saracino@uniba.it (A.S.); 2Studio Ostetrico/Nutrizionale DeaLuce, Cap 00168 Rome, Italy; laura.pennazzi@gmail.com; 3Clinic of Obstetrics & Gynaecology, University of “Aldo Moro”, Cap 70124 Bari, Italy; miriana729@gmail.com (M.B.); amerigo.vitagliano@uniba.it (A.V.); ettore.cicinelli@uniba.it (E.C.); 4Unit of Endocrinology, Department of Translational Medicine and Surgery, Università Cattolica del Sacro Cuore, Fondazione Policlinico “A. Gemelli” IRCCS, Cap 00168 Rome, Italy; pietro.locantore@policlinicogemelli.it

**Keywords:** antibiotic treatment, chronic endometritis, fertility, multidrug resistance, persistent chronic endometritis

## Abstract

Chronic endometritis (CE) is a persistent inflammatory condition of the endometrium characterized by abnormal infiltration of plasma cells into the endometrial stroma. Frequently associated with repeated implantation failure, recurrent pregnancy loss, and infertility, CE significantly impacts women’s health, contributing to conditions such as abnormal uterine bleeding and endometriosis. Treatment typically involves antibiotic therapy; however, the efficacy of these treatments is increasingly compromised by the rise of antimicrobial resistance (AMR). This paper examines the critical links between AMR and CE, proposing strategies to enhance clinical management and optimize treatment outcomes.

## 1. Introduction

Chronic endometritis (CE) is defined as a persistent local inflammatory condition of the endometrium, characterized by an abnormal infiltration of plasma cells within the endometrial stroma [1]. CE is frequently detected in cases of repeated implantation failure and recurrent pregnancy loss, and it may lead to several conditions that significantly impact a woman’s health and well-being, including infertility, recurrent miscarriages, abnormal uterine bleeding and endometriosis [2,3]. Treatment consists of antibiotic treatment that can be guided on the results of endometrial cultures and antibiogram or more frequently based on standard therapy with first-line antibiotics like doxycycline and quinolones. However, recent research has raised concerns about the failure of first-line antibiotics in treating chronic endometritis, potentially due to links with antimicrobial resistance (AMR) [4]. This reduces quality of life and decreases the likelihood of embryo implantation, posing substantial challenges to effective treatment and management. This issue is particularly pressing in the context of global antibiotic resistance, where Italy and Greece have some of the poorest rankings in AMR [5]. Moreover, patients with CE often have underlying conditions that may expose them to multiple courses of antibiotic therapy (e.g., chronic cystitis, vaginitis, etc.), which increases the risk of multidrug-resistant pathogen development.

Additionally, medical malpractice, including the over-prescription of antibiotics, exacerbates the rise of multi-resistant infections [6]. As a result, women of childbearing age are increasingly at risk of endometritis caused by multi-resistant pathogens [7].

This may provide a significant challenge to women’s health, similarly to the impacts of SARS-CoV infection in pregnancy [8,9]. This literature review aims to examine the increasing correlations between antimicrobial resistance (AMR) and chronic endometritis (CE), providing a framework for clinicians to enhance management and treatment strategies. By exploring the interaction between resistant pathogens and the persistent inflammation in CE, this work seeks to address knowledge gaps and support the development of evidence-based approaches for the effective treatment and long-term resolution of CE in the context of growing AMR challenges.

## 2. Methods

We searched PubMed, Scopus, Google Scholar, EMBASE, Cochrane Library, and WHO websites (http://www.who.int) for literature addressing chronic endometritis, published up to August 2024. The search strategy included terms such as: “Chronic endometritis [tiab]”, “CE [mh]”, “infertility”, “Antimicrobial resistance [tiab]”, “CE, resistance [tiab]” and “CE, treatment failure [tiab]”, “Chronic endometritis, resistance [tiab]”. All studies addressing epidemiology, physiopathology, clinical characteristics, screening and diagnosis, therapy, and management were included. We included both clinical studies and systematic reviews that offered insights into the influence of antimicrobial resistance on chronic endometritis, with a particular focus on treatment outcomes, diagnostic methodologies, and emerging trends in the management of resistant strains.

## 3. Pathogenesis and Risk Factors of Chronic Endometritis and Implication in Antibiotic Resistance

CE is characterized by specific cellular alterations [1]. Compared with normal endometrium, women with CE exhibit an increased number of B lymphocytes, which infiltrate both the functional and basal layers of the endometrium. In these areas, B lymphocytes breach the glandular epithelium and enter the glandular lumens. Some of these infiltrating endometrial B cells may locally mature into endometrial stromal plasma cells (ESPCs) [10]. These ESPCs produce elevated levels of various immunoglobulin (Ig) subclasses, particularly IgG2 [11]. This heightened mucosal antibody production in CE may negatively affect embryo implantation [12]. The inflammatory environment is often due to imbalances in the endometrial microbiota, typically resulting from ascending microbial infections from the lower to upper genital tract [13]. Antibiotic therapy has shown up to a 100% reduction in ESPCs in CE patients [14].

Fluid hysteroscopy is a practical technique, widely acknowledged as a diagnostic tool with high specificity, as demonstrated in various studies [15,16].

In 2019, following a systematic review of previous studies and consensus reached through the Delphi poll, the International Working Group for the Standardization of Chronic Endometritis Diagnosis established specific criteria for diagnosing CE [17]. A histological diagnosis requires the presence of 1 to 5 ESPCs per high-power field or clusters of fewer than 20 ESPCs, as identified by CD138 staining. Additionally, the following hysteroscopic findings during the follicular phase of the menstrual cycle are diagnostic of CE [18]:Endometrial micro-polyposis (1–2 mm protrusions from the endometrial surface) [19].Stromal edema, which causes the endometrium to appear thick and pale during the follicular phase rather than the secretory phase [20].Focal reddened areas of the endometrium with sharp, irregular borders [21].Large regions of hyperemic endometrium with white central points [19].Focal hyperemia.

Recent studies, such as the ARCHIPELAGO study [21], highlight the potential of deep learning models to develop predictive tools based on hysteroscopic findings, emphasizing the need for further research to refine the correlation between hysteroscopic and histopathological results to improve diagnostic accuracy and clinical outcomes.

Infections of the upper genital tract are attributed to several factors:Recurrent cystitis: Women with a history of recurrent urinary tract infections (UTIs) face an increased risk of ascending infections due to the proximity of the urethra to the vagina and cervix [22].Vaginal transmission of intestinal germs: Intestinal bacteria, particularly Escherichia coli, Klebsiella spp., and Enterococcus spp., can colonize the vaginal and perineal areas, leading to uterine infections [23]. Poor hygiene, fecal contamination during intercourse, or improper wiping techniques after bowel movements can facilitate this transmission. Stress is also strongly associated with increased translocation of intestinal bacteria to the urogenital tract, raising the risk of CE [24]. See Figure 1.

Chlamydia trachomatis remains the leading sexually transmitted infection in Europe, with the highest notification rates in women aged 20–24 years in 2022. The rate increased by 18% compared with 2021, according to the ECDC Epidemiological Report for 2022 [25]. This infection causes considerable acute morbidity and long-term complications, including infertility and adverse pregnancy outcomes [26]. Unlike acute endometritis (AE), Chlamydia trachomatis and Neisseria gonorrhoeae, which are primary pathogens in AE, play a limited role in the pathogenesis of CE, as shown in several studies [12,26,27].

## 4. Antibiotic Resistance and MDR in Chronic Endometritis

The emergence of AMR is a complex, multifactorial issue that poses a significant global burden, affecting health, economic, and social dimensions. Each year, an estimated 7.7 million deaths are attributed to bacterial infections, with 4.95 million deaths being linked to AMR [28]. AMR refers to the ability of microorganisms to survive or grow despite the presence of antimicrobial agents commonly used to treat bacterial, fungal, viral, or protozoan infections. Multidrug resistance (MDR) occurs when microorganisms become resistant to multiple antimicrobial drugs. Typically, an organism is considered to be MDR if it resists at least one agent in three or more antimicrobial categories. The mechanisms behind MDR are similar to those of single-drug resistance but span a wider range of drugs, often due to the combination of several resistance genes. The antimicrobial and multidrug resistance (AMR/MDR) phenomenon represents a complex and evolving global challenge that critically undermines the efficacy of antibacterial therapies. This multifaceted issue arises from an interplay of genetic, ecological, and anthropogenic factors [5].

At the genetic level, resistance mechanisms are driven by chromosomal mutations; horizontal gene transfer mediated by mobile genetic elements such as plasmids, transposons, and integrons; and the dissemination of resistance determinants across diverse microbial populations [29].

Ecologically, environmental reservoirs—including soil, aquatic systems, and agricultural settings—contribute to the maintenance and propagation of resistant strains. Microbial ecosystems within humans, animals, and environmental niches interact dynamically, facilitating the emergence and persistence of resistance traits [30]. Furthermore, anthropogenic pressures, particularly the widespread misuse and overuse of antimicrobials in human medicine, agriculture, and veterinary practices, have created significant selective pressures, accelerating the proliferation and dissemination of AMR/MDR globally [31]. Bacteria in water, soil, and air can acquire resistance through exposure to already-resistant germs [6]. This phenomenon correlates with human exposure to AMR in the environment, which can occur through contact with polluted water or consumption of contaminated food, inhalation of fungal spores or through other sources harboring resistant microbes [32]. The economic burden of AMR is largely due to the prolonged hospitalizations needed to treat infections caused by resistant bacteria, which increases the risk of complications, further spreads resistance in healthcare settings, and leads to treatment failures. These failures necessitate the use of alternative antibiotics, which may be more toxic or expensive.

The rise of AMR has also affected the treatment of conditions that predispose individuals to chronic endometritis, such as bacterial vaginosis and recurrent cystitis [14]. Additionally, medical malpractice, including overprescription and inappropriate use of antibiotics, contributes to the selection of multi-resistant pathogens, ultimately reducing the efficacy of home treatments for chronic endometritis [7]. While the spread of antibiotic resistance is steadily increasing, data on its impact on chronic endometritis remain unclear. According to the European Centre for Disease Prevention and Control (ECDC) Antimicrobial Resistance Surveillance in Europe 2021–2023 report [33], *E. coli* is the most common agent of community-acquired bacteremia and urinary tract infections. Data from the WHO in 2021 on the percentage of invasive isolates resistant to fluoroquinolones (ciprofloxacin/levofloxacin/ofloxacin) reveal a significant disparity in AMR levels across regions. In 2 of the 45 countries—Finland and Norway—resistance to fluoroquinolones was below 10%, while 17 countries, including Italy, reported AMR levels of 25% or higher. Cyprus, North Macedonia, Russia, and Turkey exhibited resistance rates of 50% or more [33].

In 2021, data on resistance to third-generation cephalosporins in *E. coli* showed that 27% of the surveyed countries reported resistance rates below 10%, while 9% reported AMR levels of 50% or higher [30]. The rising global incidence of ESBL-producing *E. coli* is driven by both community-acquired and healthcare-associated infections [34].

Resistance to third-generation cephalosporins in *K. pneumoniae* has significantly increased across the WHO European Region. In 2021, 42% of countries reported resistance rates of 50% or higher. *K. pneumoniae* also exhibited a higher prevalence of carbapenem resistance compared with *E. coli* [6]. This increase in resistance is concerning, as frequent and repeated antibiotic use selects for resistant strains, including ESBL-producing or carbapenem-resistant *K. pneumoniae*. These resistant strains pose substantial challenges, particularly in the context of relapsing urinary tract infections (UTIs) due to inappropriate prescriptions or insufficient microbiological data to guide therapy. These challenges often require a shift to intravenous (IV) antibiotics, resulting in increased healthcare costs, longer hospital stays, and higher mortality rates [33].

In recent years, *Enterococcus* spp. has gained attention as a cause of nosocomial infections due to its ability to cause MDR infections [35]. *Enterococci* naturally exhibit resistance to several classes of antimicrobials, and any additional AMR further restricts treatment options. The WHO has classified vancomycin-resistant *E. faecium* as a high-priority pathogen on its global list of antibiotic-resistant bacteria, underscoring the limited availability of effective treatments [33]. Although high-level gentamicin resistance in *E. faecalis* remains stable according to European Antimicrobial Resistance Surveillance Network (EARS-Net) data, the persistence of high resistance emphasizes the ongoing challenge of managing antimicrobial-resistant *Enterococci*, which cause significant healthcare-associated infections in Europe [33]. By 2021, 17.2% of *E. faecium* isolates were vancomycin-resistant. Additionally, nearly one-third of all *E. faecalis* isolates reported to EARS-Net showed high-level resistance to gentamicin, and 45.2% of *E. faecium* isolates were resistant to two antimicrobial groups [28,35]. A staggering 93.0% of *E. faecium* isolates were resistant to at least one antimicrobial group under surveillance (aminopenicillins, gentamicin, and vancomycin) [28].

In 2021, fluoroquinolone resistance levels were generally lower in the northern and western regions of the WHO European Region but higher in the southern and eastern regions. As a result, first-line antibiotics, which were highly effective in treating chronic endometritis just a few years ago, may now be rendered ineffective. This finding is substantiated not only by the latest ECDC report [28], which provides a comprehensive analysis of current epidemiological trends but also by a recent meta-analysis conducted by Naghavi et al. [32], which represents the first comprehensive assessment of the global burden of antimicrobial resistance (AMR) from 1990 to 2021 and highlights the interplay between epidemiological patterns and various environmental and climatic conditions, offering a detailed understanding of the observed outcomes and their potential future implications.

Therefore, studies on antibiotic resistance in pathogens causing endometritis are crucial for revising therapeutic recommendations and tailoring treatments to specific pathogens and their resistance profiles.

A meta-analysis conducted by Kato and colleagues [36] on pregnancy outcomes in CE highlighted that standard antibiotic treatments did not improve implantation rates, illustrating the risk of failure with first-line regimens in the case of MDR infections [37]. As shown in Table 1, according to the CDC, first-line treatment for CE involves the use of doxycycline, with metronidazole/ciprofloxacin serving as second-line treatments [38]. However, recent studies indicate growing resistance to these first-line treatments [39]

Moreover, evidence suggests that administering corticosteroids in combination with antibiotics may enhance reproductive outcomes.

Corticosteroids, with their anti-inflammatory properties, can reduce immune-mediated damage and improve tissue receptivity, while antibiotics address underlying infections, creating a more favorable environment for successful implantation and pregnancy [40,41].

Subsequent studies have shown that another quinolone less commonly used in routine clinical practice, moxifloxacin, exhibits superior activity against the pathogens responsible for CE, surpassing both ciprofloxacin and metronidazole. This makes moxifloxacin a promising therapeutic option for managing multidrug-resistant chronic endometritis (MDR-CE) [1]. In Table 2, we summarize the characteristics of several studies focused on the treatment of CE, including information on the authors, country, study design, etiological agents, treatments used, treatment durations, and a comparison with the most recent antimicrobial resistance (AMR) data from the CDC/WHO Antimicrobial Resistance Surveillance in Europe 2023–2021 report [33]. Based on this data, we offer a personal perspective by stratifying the risk of treatment failure.

The studies conducted in various countries including Italy, Japan, China, the USA, Turkey, and Argentina, vary in sample size and study design (retrospective and prospective) [42] describe the different etiological agents responsible for CE, such as *Escherichia coli*, *Streptococci*, *Enterococcus faecalis*, *Ureaplasma*, and *Mycoplasma*. The treatments commonly involve antibiotics such as ciprofloxacin, amoxicillin/clavulanate, doxycycline, levofloxacin, and combination therapies with metronidazole. We evaluate the potential risk of treatment failure by comparing the study results with the CDC/WHO Antimicrobial Resistance Surveillance in Europe 2023–2021 data.

The risk of treatment failure is stratified based on two critical factors: the antimicrobial resistance rate and the type of pathogen involved. These factors are essential in predicting the success or failure of therapy. We define the risks as follows:Low risk:AMR rate: resistance rates for etiological agents are below 10% for first-line antibiotics.Pathogen type: well-known and easily treatable pathogens with low virulence and minimal resistance (e.g., *Streptococcus agalactiae*).
Moderate risk:AMR rate: resistance rates range from 10% to 30%.Pathogen type: pathogens with partial resistance to commonly used antibiotics (e.g., *Escherichia coli* resistant to fluoroquinolones).High (severe) risk:AMR rate: resistance rates exceed 30%, particularly for multidrug-resistant bacteria (e.g., *E. coli* or *Klebsiella pneumoniae*).Pathogen type: multidrug-resistant pathogens that are difficult to treat, such as *Klebsiella pneumoniae* resistant to carbapenems or ESBL-producing *E. coli*.


This assessment is based on CDC/WHO AMR surveillance data, underscoring the growing importance of updated antimicrobial stewardship practices to combat resistance trends and improve clinical outcomes in CE treatment.

**Table 2 microorganisms-13-00197-t002:** Characteristics of the studies about CE treatment.

Authors	Country and Year	Patients with CE	Study Design	Etiological Agents	Treatment Procedure	Duration of Treatment	CDC/WHO Antimicrobial Resistance Surveillance in Europe 2023–2021 Data	Risk of Failure
Cicinelli et al. [43]	Italy, 2021	128	Retrospective study	*Escherichia coli* 38/128 *Streptococci* 31/128 *Staphylococci* 2/128 *Enterococcus faecalis* 33/128 *Klebsiella pneumoniae* 2/128 *Ureaplasma* 36/128 Yeast 2/128	Repeated course (up to three times)		30% of *E. coli* isolates show AMR phenotype to aminopenicillins, 5.4% show AMR phenotype to Aminopenicillins + Fluoroquinolones	High
					Ciprofloxacin 500 mg twice a day (if Gram negative)	10 days		
					Amoxicillin/Clavulanate 1 g twice a day (if Gram-positive)	8 days		
					Josamycin 1 g twice a day (if mycoplasma and *U. urealyticum*) PLUS Minocycline 100 mg twice a day (if persistent)	12 days		
Kitaya et al. [39]	Japan, 2017	142	Prospective study	*Enterococcus* 15/142 (10.6) *Escherichia coli* 14/142 (9.9) *Ureaplasma parvum* 14/46 (30.4) *Mycoplasma hominis* 8/46 (17.4) *Streptococcus agalactiae* 8/142 (5.6) *Corynebacterium* 10/142 (7.0) *Staphylococcus aureus* 7/142 (4.9) *Lactobacillus* 7/142 (4.9) *Ureaplasma urealyticum* 6/46 (13.0) *Staphylococcus saprophyticus* 4/142 (2.8) *Mycoplasma genitalium* 4/46 (8.7) *Streptococcus pyogenes* 3/142 (2.1) *Klebsiella pneumoniae* 2/142 (1.4) *Staphylococcus epidermidis* 1/142 (0.7) *Chlamydia trachomatis* 2/142 (1.4)	Repeated course (up to two times)		21.6% of *E. coli* isolates show AMR phenotype to fluoroquinolones, including Ciprofloxacin 33.6% of K. pneumoniae isolates with AMR phenotype to fluoroquinolones, including ciprofloxacin	Moderate
					Doxycycline 100 mg twice a day	14 days		
					Metronidazole 250 mg twice a day PLUS Ciprofloxacin hydrochloride 200 mg twice a day (if resistance to doxycycline)	14 days		
Yang et al. [21]	China, 2014	88	Retrospective study	no data	Single course	14 days		
					Levofloxacin 500 mg once a day PLUS Metronidazole 1 g once a day			
Johnston-MacAnanny et al. [44]	USA, 2009	43	Retrospective study	no data	Doxycycline 100 mg twice a day	14 days		
					Ciprofloxacin PLUS metronidazole 500 mg twice a day, respectively	14 days		
Xiong et al. [45]	China, 2021	26	Retrospective study	no data	Doxycycline 100 mg twice a day	14 days		
					Levofloxacin 200 mg twice a day PLUS Metronidazole 500 mg three times a day	14 days		
Demirdag et al. [46]	Turkey, 2021	129	Retrospective study	no data	Single course			
					Ciprofloxacin 500 mg twice a day PLUS Ornidazole 500 mg twice a day	14 days		
Tersoglio et al. [47]	Argentina, 2015	14	Prospective study	no data	Doxycycline 100 mg twice a day	14 days		
					Metronidazole 1 g once a day PLUS Ciprofloxacin 1 g once a day (if culture negative)	14 days		
					Linezolid 600 mg once a day (if persistent)	10 days		
Cicinelli et al. [4]	Italy, 2015	61	Retrospective study	*Enterococcus faecalis* 16/61 (33) *Mycoplasma*/*Ureaplasma* 14/61 (30) *Escherichia coli* 11/61 (23) *Streptococcus agalactiae* 5–61 (10) *Chlamydia* 4/61 (8) *Streptococcus bovis* 2/61 (4) *Candida* 1/61 (2) *Klebsiella pneumoniae* 1/61 (2) *Staphylococcus epidermidis* 1/61 (2) *Staphylococcus aureus* 1/61 (2) *Streptococcus milleri* 1/61 (2)	First-line therapy: Ciprofloxacin 500 mg twice a day	10 days	21.6% of *E. coli* isolates show AMR phenotype to fluoroquinolones, including Ciprofloxacin 33.6% of *K. pneumoniae* isolates with AMR phenotype to fluoroquinolones, including ciprofloxacin 4.7% of *S. aureus* isolates with AMR phenotype to fluoroquinolones	Moderate
					Amoxicillin/Clavulanate 1 g twice a day (In case of gram-positive bacteria)	8 days		
					Josamycin 1 g twice a day in case of *Mycoplasma* spp. and *U. urealyticum*	12 days		
					Minocycline 100 mg twice a day (in case of persistence)	12 days		

## 5. Nutritional Role in Chronic Endometritis

Traditionally, the uterine microenvironment was considered sterile. However, recent genomic research, including the discovery of 16S rRNA in the uterine compartment, has revealed the presence of bacteria in the uterus [48]. The female reproductive tract hosts distinct microbial communities in the vagina, cervix, uterus, and fallopian tubes, and alterations in the uterine microbiota can play a crucial role in uterine-related pathologies and impair female fertility [27,42].

As part of the mucosal immune system, the endometrium provides an immunologically suitable niche for the microbiota, with a potential role in modulating inflammatory and immune responses [49]. The mucosal immune system in the endometrium features genital epithelial cells expressing pattern recognition receptors (PRRs) like Toll-like receptors (TLRs) and NOD-like receptors, which are essential for defending against pathogen invasion and facilitating tissue adaptation and reproductive success. The molecular functions of the endometrial microbiota have been linked to metabolism, genetic information processing, immune system regulation, and cellular signaling processes [43,44]. The intestinal microbiota and female genital tract microbiota represent highly complex, interconnected ecosystems [45].

The crosstalk between these two systems is vital for maintaining physiological, immunological, and metabolic homeostasis [46]. The human endometrium undergoes cyclic processes such as shedding, repair, regeneration, and remodeling. As part of this dynamic system, the mucosa serves as a protective barrier, not only guarding against pathogens but also promoting immune tolerance, which is essential for successful pregnancy [44].

Endometrial receptivity is key to embryo implantation, and immunological tolerance to fetal antigens, as well as tightly regulated inflammatory mediators, are fundamental in this context [47,50].

Failures in implantation during assisted reproduction are often due to low-quality embryos and poor endometrial receptivity, both of which are influenced by the cellular immune response and microbiota composition [47,51].

A proper diet and the use of probiotics can positively influence the composition of the intestinal microbiota, improve intestinal integrity, and help maintain or restore normal vaginal microbiota [52].

Numerous studies demonstrate that diet is a key modifiable factor influencing intestinal microflora composition [50]. The Mediterranean diet, rich in fruits, vegetables, whole grains, healthy fats (e.g., olive oil), and lean proteins, provides essential nutrients and antioxidants that contribute to an overall anti-inflammatory effect and enhanced immune function [47]. This dietary approach promotes the growth of beneficial bacteria, particularly Lactobacillus species, which are crucial for maintaining vaginal pH and preventing pathogenic overgrowth. High fiber content in the diet supports gut health, which is increasingly understood to have a bidirectional relationship with vaginal health via the gut-vaginal axis [53].

Pre- and probiotic supplements, along with a balanced diet low in fats and rich in folates, antioxidants, and vitamins (E, C, A, and D), can help maintain the barrier function of the intestinal mucosa and reduce the proliferation of pathogenic microorganisms [47,50]. An imbalanced diet, marked by high energy density and low intake of essential micronutrients, may increase the risk of developing bacterial vaginosis (BV) [47]. In addition to direct microbial transmission, the gut microbiota, via the estrobolome—the assemblage of gut bacteria that can metabolize estrogens—indirectly influences hormone levels that impact the composition of the genital microbiota and the health of the reproductive tract [51,53,54].

Dysbiosis in the gut or vaginal microbiota has been linked to several reproductive tract disorders, including BV, cervical and endometrial cancer, polycystic ovary syndrome (PCOS), postmenopausal syndrome, endometriosis, endometritis, and uterine fibroids (UFs) [50].

Probiotics have been shown to support vaginal health by competitively excluding pathogens, producing bacteriocins, and reinforcing the vaginal microbiota’s natural protective functions [50]. This helps prevent and treat infections such as vulvovaginal candidiasis (VVC) and BV [46]. Bastani et al. [55] confirmed the effectiveness of probiotics, particularly *L. acidophilus*, *L. rhamnosus* GR-1, and *L. fermentum* RC-14, in restoring normal urogenital flora and preventing BV recurrence. Prebiotics, naturally present in foods like garlic, chicory, artichokes, and bananas, selectively stimulate the growth of beneficial bacteria in the colon, supporting overall health by modulating the microbial ecosystem [47]. Personalized nutritional interventions that address nutrient deficiencies and reinforce protective factors, such as prebiotic fibers, can significantly contribute to optimizing endometrial health and female fertility [56,57].

## 6. Conclusions

CE remains as an underdiagnosed and poorly known pathology, but it is a frequent cause of infertility, significantly impacting both the quality of life and the patient’s right to parenthood. This paper highlights the growing global issue of antibiotic resistance and the emergence of multidrug-resistant pathogens causing chronic endometritis, which increases the risk of failure in first-line or empirical treatments. In addition to global AMR risk factors, patients with endometritis frequently have conditions that require high antibiotic use (e.g., chronic cystitis, vaginitis), making them more vulnerable to infections caused by multidrug-resistant organisms.

Limitations of this study should be acknowledged: The reliance on existing literature and secondary data may introduce biases or inconsistencies due to variations in study design, sample populations, and diagnostic criteria across sources. Additionally, the complex interactions between antimicrobial resistance (AMR) and chronic endometritis (CE) are influenced by several factors, such as patient comorbidities, hormonal variations, and environmental influences, which were not fully explored in this work. Furthermore, the heterogeneity in the methodologies used to detect and characterize microbial pathogens and their resistance profiles may limit the generalizability of conclusions. Future research incorporating standardized diagnostic protocols, larger and more diverse patient cohorts, and longitudinal designs would be essential to address these limitations and deepen the understanding of AMR’s role in CE. A comprehensive exploration of AMR/MDR requires a systematic approach, addressing specific subtopics such as molecular resistance mechanisms, gene dissemination pathways, and their impact on resistance dynamics.

Effective chronic endometritis (CE) management necessitates a coordinated, multidisciplinary approach involving infectious disease specialists, gynecologists, nutritionists, and other relevant professionals. This approach should include the establishment of surveillance programs, the advancement of diagnostic techniques, and the creation of a unified strategy to improve both our understanding of AMR/MDR and therapeutic outcomes in CE.

## Figures and Tables

**Figure 1 microorganisms-13-00197-f001:**
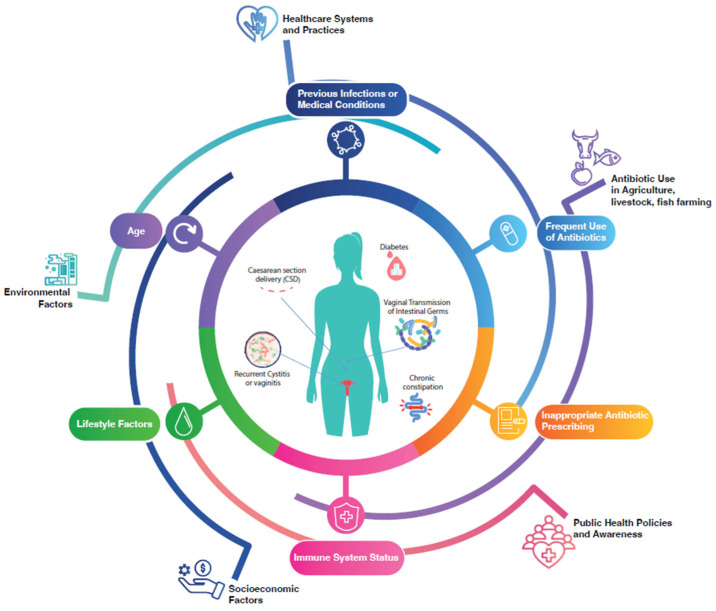
Extrinsic and intrinsic risk factors for CE.

**Table 1 microorganisms-13-00197-t001:** Antibiotic regimes recommended by CDC for CE [38].

Line	Antibiotic	Dosage	Duration of Treatment
1°	Doxycycline	100 mg orally twice daily	14 days
2°	Metronidazole	500 mg orally daily	14 days
PLUS			
	Ciprofloxacin	400 mg orally daily	14 days

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
