# Peer review of "Chronic Endometritis and Antimicrobial Resistance: Towards a Multidrug-Resistant Endometritis? An Expert Opinion"

_microorganisms, 2025, doi:10.3390/microorganisms13010197_

Round 1
Reviewer 1 Report
Comments and Suggestions for Authors
This literature review discusses through an expert opinion the association between chronic endometritis and antimicrobial resistance. The review is interesting, has a global interest and provides useful information. The following is simple comments should be addressed before acceptance
- Key words should be arranged alphabetically and differ from those mentioned in the title to expand the visibility of the article in the search engine.
- Line 31: replace “consists on” with “consists of”.
- Line 130: please define ECDC, and invert 2023-2021.
- Line 158: define EARS.
- Line 166-168: “In 2021, fluoroquinolone resistance levels were generally lower in the northern and western regions of the WHO European Region but higher in the southern and eastern regions” ; can the authors provide an explanation to this difference.
- Line 176: Define CDC.
- Line 179: “chronic endometritis (CE)” this word is already abbreviated throughout the manuscript; please use the abbreviation only.
- Please add a paragraph discussing the genetic background of antimicrobial and multidrug resistance.
-
Author Response
To Microorganism Editor,
We have appreciated the feedback on our manuscript “Chronic Endometritis and Antimicrobial Resistance: Towards a Multi-Drug-Resistant Endometritis? An Expert Opinion”.
We have considered all the valuable suggestions made by the referees and implemented the text. We have also satisfied the technical requirements according to the journal guidelines. Modifications have been highlighted using the “track changes” feature. We believe that the revision proposed by the reviewers, and further implemented in the text, contributed to improving the manuscript.
Thus, we kindly ask you to reconsider the manuscript for publication. Please find a point-by-point response to the referees’ comments below.
Best regards,
Dr. Luisa Frallonardo
Reviewer1
This literature review discusses through an expert opinion the association between chronic endometritis and antimicrobial resistance. The review is interesting, has a global interest and provides useful information. The following is simple comments should be addressed before acceptance
- Key words should be arranged alphabetically and differ from those mentioned in the title to expand the visibility of the article in the search engine.
Response: Thank you for your observation. We modified Key words following your suggestion.
- Line 31: replace “consists on” with “consists of”.
- Line 130: please define ECDC, and invert 2023-2021.
- Line 158: define EARS.
Response: Thank you for your observations.We correct the spell in the manuscript.
- Line 166-168: “In 2021, fluoroquinolone resistance levels were generally lower in the northern and western regions of the WHO European Region but higher in the southern and eastern regions” ; can the authors provide an explanation to this difference.
Response: Thank you very much for your observation. We implemented the paragraph as follows:
”This finding is substantiated not only by the latest ECDC report [46], which provides a comprehensive analysis of current epidemiological trends but also by a recent meta-analysis conducted by Naghavi et al [73] which represents the first comprehensive assessment of the global burden of antimicrobial resistance (AMR) from 1990 to 2021 and highlights the interplay between epidemiological patterns and various environmental and climatic conditions, offering a detailed understanding of the observed outcomes and their potential future implications.”
- Line 176: Define CDC.
- Line 179: “chronic endometritis (CE)” this word is already abbreviated throughout the manuscript; please use the abbreviation only.
Response: Thank you. We corrected it following your suggestion
- Please add a paragraph discussing the genetic background of antimicrobial and multidrug resistance.
Response: Thank you for your suggestion. We implemented as follows
“The antimicrobial and multidrug resistance (AMR/MDR) phenomenon represents a complex and evolving global challenge that critically undermines the efficacy of antibacterial therapies. This multifaceted issue arises from an interplay of genetic, ecological, and anthropogenic factors. [5]At the genetic level, resistance mechanisms are driven by chromosomal mutations, horizontal gene transfer mediated by mobile genetic elements such as plasmids, transposons, and integrons, and the dissemination of resistance determinants across diverse microbial populations. Ecologically, environmental reservoirs—including soil, aquatic systems, and agricultural settings—contribute to the maintenance and propagation of resistant strains. Microbial ecosystems within humans, animals, and environmental niches interact dynamically, facilitating the emergence and persistence of resistance traits. Furthermore, anthropogenic pressures, particularly the widespread misuse and overuse of antimicrobials in human medicine, agriculture, and veterinary practices, have created significant selective pressures, accelerating the proliferation and dissemination of AMR/MDR globally.”[74]
Reviewer 2 Report
Comments and Suggestions for Authors
Review Report
20.11.2024
I was invited to review the manuscript:
Chronic Endometritis and Antimicrobial Resistance: Towards a Multi-Drug-Resistant Endometritis? An Expert Opinion
The topic of the manuscript is interesting. Generally, the review article enclosed the main research domains of these subject
1. The methodology of review should be detailed. PRISMA diagram should be presented.
2. The content of sections 3,4 and5 should be presented as subsections of the Results and Discussions sections, according to the template of the manuscript.
3. The statement: “A proper diet and the use of probiotics can positively influence the composition of the intestinal microbiota, improve intestinal integrity, and help maintain or restore normal vaginal microbiota.” (R 49-50) should be explained. How does the intestinal microbiota influence the vaginal microbiota”.
4. Limits of the present studies and future research perspectives should be added.
Sincerely,
Author Response
To Microorganism Editor,
We have appreciated the feedback on our manuscript “Chronic Endometritis and Antimicrobial Resistance: Towards a Multi-Drug-Resistant Endometritis? An Expert Opinion”.
We have considered all the valuable suggestions made by the referees and implemented the text. We have also satisfied the technical requirements according to the journal guidelines. Modifications have been highlighted using the “track changes” feature. We believe that the revision proposed by the reviewers, and further implemented in the text, contributed to improving the manuscript.
Thus, we kindly ask you to reconsider the manuscript for publication. Please find a point-by-point response to the referees’ comments below.
Best regards,
Dr. Luisa Frallonardo
Reviewer2
I was invited to review the manuscript:
Chronic Endometritis and Antimicrobial Resistance: Towards a Multi-Drug-Resistant Endometritis? An Expert OpinionThe topic of the manuscript is interesting. Generally, the review article enclosed the main research domains of these subject
- The methodology of review should be detailed. PRISMA diagram should be presented.
Response: Thank you.The PRISMA diagram was not employed in this study, as it does not constitute a systematic review. Nonetheless, we acknowledge the potential applicability of this framework and may consider this methodology in future studies,
Table 2 already include information regarding the studies involved.
- The content of sections 3,4 and 5 should be presented as subsections of the Results and Discussions sections, according to the template of the manuscript.
Response: Thank you. We modified the sections by inserting the subsections.
- The statement: “A proper diet and the use of probiotics can positively influence the composition of the intestinal microbiota, improve intestinal integrity, and help maintain or restore normal vaginal microbiota.” (R 49-50) should be explained. How does the intestinal microbiota influence the vaginal microbiota”.
Response: Thank you for your observation. We implemented as follows:
“In addition to direct microbial transmission, the gut microbiota, via the estro-bolome—the assemblage of gut bacteria that can metabolize estrogens—indirectly in-fluences hormone levels that impact the composition of the genital microbiota and the health of the reproductive tract.[38][17]
Dysbiosis in the gut or vaginal microbiota has been linked to several reproductive tract disorders, including BV, cervical and endometrial cancer, polycystic ovary syn-drome (PCOS), postmenopausal syndrome, endometriosis, endometritis, and uterine fibroids (UFs). [76]”
- Limits of the present studies and future research perspectives should be added.
Response: Thank you for your observation. We implemented as follows:
“Limitations of this study should be acknowledged: the reliance on existing literature and secondary data may introduce biases or inconsistencies due to variations in study design, sample populations, and diagnostic criteria across sources. Additionally, the complex interactions between antimicrobial resistance (AMR) and chronic endo-metritis (CE) are influenced by several factors, such as patient comorbidities, hormonal variations, and environmental influences, which were not fully explored in this work. Furthermore, the heterogeneity in the methodologies used to detect and characterize microbial pathogens and their resistance profiles may limit the generalizability of conclusions. Future research incorporating standardized diagnostic protocols, larger and more diverse patient cohorts, and longitudinal designs would be essential to address these limitations and deepen the understanding of AMR's role in CE. A comprehensive exploration of AMR/MDR requires a systematic approach, addressing specific subtop-ics such as molecular resistance mechanisms, gene dissemination pathways, and their impact on resistance dynamics.
Effective chronic endometritis (CE) management necessitates a coordinated, multidisciplinary approach involving infectious disease specialists, gynecologists, nutritionists, and other relevant professionals. This approach should include the establishment of surveillance programs, the advancement of diagnostic techniques, and a unified strategy to improve both our understanding of AMR/MDR and therapeutic outcomes in CE”
Reviewer 3 Report
Comments and Suggestions for Authors
This review is a mini but comprehensive work on the topic of Chronic Endometritis and Antimicrobial Resistance. There are some typing errors throughout the article. For example, page 2, line 53, “Antimicrobial resistance [tiab]”, “CE, resistence", the later resistance is a typing error. Please check this article again and correct.
Comments on the Quality of English Language
This review is a mini but comprehensive work on the topic of Chronic Endometritis and Antimicrobial Resistance. There are some typing errors throughout the article. For example, page 2, line 53, “Antimicrobial resistance [tiab]”, “CE, resistence", the later resistance is a typing error. Please check this article again and correct.
Author Response
To Microorganism Editor,
We have appreciated the feedback on our manuscript “Chronic Endometritis and Antimicrobial Resistance: Towards a Multi-Drug-Resistant Endometritis? An Expert Opinion”.
We have considered all the valuable suggestions made by the referees and implemented the text. We have also satisfied the technical requirements according to the journal guidelines. Modifications have been highlighted using the “track changes” feature. We believe that the revision proposed by the reviewers, and further implemented in the text, contributed to improving the manuscript.
Thus, we kindly ask you to reconsider the manuscript for publication. Please find a point-by-point response to the referees’ comments below.
Best regards,
Dr. Luisa Frallonardo
Reviewer3
This review is a mini but comprehensive work on the topic of Chronic Endometritis and Antimicrobial Resistance. There are some typing errors throughout the article. For example, page 2, line 53, “Antimicrobial resistance [tiab]”, “CE, resistence", the later resistance is a typing error. Please check this article again and correct.
Response: Thank you for your observations. We correct the spelling in the manuscript.
Reviewer 4 Report
Comments and Suggestions for Authors
Dear author’s,
I was pleased to review your paper entitled “ Chronic Endometritis and Antimicrobial Resistance: towards a Multi-drug-resistant endometritis? An expert opinion” and i have the following comment’s:
1. The abstract should be revised in order to explain that ythe study is a literature review.
2. Please explain in detail the aim of this paper.
3. The methodology should contain more information in order to explain the information.
4. A flowchart with the studies involved could be interesting.
5. Why do you choose this subject? What new info do you want to highlight?
6. It is mandatory to highlight the role of hysteroscopic examination in this pathology.
7. Minor English and punctuation edits are necessary.
Author Response
To Microorganism Editor,
We have appreciated the feedback on our manuscript “Chronic Endometritis and Antimicrobial Resistance: Towards a Multi-Drug-Resistant Endometritis? An Expert Opinion”.
We have considered all the valuable suggestions made by the referees and implemented the text. We have also satisfied the technical requirements according to the journal guidelines. Modifications have been highlighted using the “track changes” feature. We believe that the revision proposed by the reviewers, and further implemented in the text, contributed to improving the manuscript.
Thus, we kindly ask you to reconsider the manuscript for publication. Please find a point-by-point response to the referees’ comments below.
Best regards,
Dr. Luisa Frallonardo
Reviewer 4
I was pleased to review your paper entitled “ Chronic Endometritis and Antimicrobial Resistance: towards a Multi-drug-resistant endometritis? An expert opinion” and i have the following comment’s:
- The abstract should be revised in order to explain that the study is a literature review.
Response: Thank you for your suggestion. We implemented as follows:
“Given these considerations, this literature review paper aims to highlight the growing correlations between AMR and chronic endometritis, offering a framework for clinicians to improve the management and treatment of CE.”
- Please explain in detail the aim of this paper.
Response: Thank you for your question. We implemented the introduction as follows:
This literature review aims to examine the increasing correlations between antimicrobial resistance (AMR) and chronic endometritis (CE), providing a framework for clinicians to enhance management and treatment strategies. By exploring the interaction between resistant pathogens and the persistent inflammation in CE, this work seeks to address knowledge gaps and support the development of evidence-based approaches for effective treatment and long-term resolution of CE amid the growing AMR challenge
3.The methodology should contain more information in order to explain the information.
Response: Thank you for your question. We implemented the introduction as follows:
We included both clinical studies and systematic reviews that offered insights into the influence of antimicrobial resistance on chronic endometritis, with a particular focus on treatment outcomes, diagnostic methodologies, and emerging trends in the management of resistant strains
4..A flowchart with the studies involved could be interesting.
Response: Thank you for your suggestion. Table 2 already includes information regarding the studies involved.
- Why do you choose this subject? What new info do you want to highlight?
Response: Thank you for your question.
This paper highlights the growing global issue of antibiotic resistance (AMR) and the emergence of multidrug-resistant pathogens causing chronic endometritis, which increases the risk of failure in first-line treatments. Patients with endometritis often have conditions requiring high antibiotic use, such as chronic cystitis and vaginitis, making them more vulnerable to infections by multidrug-resistant organisms
6.It is mandatory to highlight the role of hysteroscopic examination in this pathology
Response: Thank you for your question. We implemented the introduction as follows:
"Fluid hysteroscopy is a practical technique, widely acknowledged as a diagnostic tool with high specificity, as demonstrated in various studies.[13][20]. In 2019, following a systematic review of previous studies and consensus reached through the Delphi poll, the International Working Group for the Standardization of Chronic Endometritis Diagnosis established specific criteria for diagnosing CE.
Recent studies, such as the ARCHIPELAGO study[77], highlight the potential of deep learning models to develop predictive tools based on hysteroscopic findings, emphasizing the need for further research to refine the correlation between hysteroscopic and histopathological results to improve diagnostic accuracy and clinical outcomes.”
- Minor English and punctuation edits are necessary.
Response: Thank you for your observations. We correct the spelling in the manuscript.
Round 2
Reviewer 2 Report
Comments and Suggestions for Authors
Minor revisions of the format are required.
The manuscript is improved and could be published.
Author Response
Thank you very much for your suggestions and your approval